# Association between the Upper Quarter Dynamic Balance, Anthropometrics, Kinematics, and Swimming Speed

**DOI:** 10.3390/jfmk8030096

**Published:** 2023-07-10

**Authors:** Raul F. Bartolomeu, Tatiana Sampaio, João P. Oliveira, Tiago M. Barbosa, Jorge E. Morais

**Affiliations:** 1Department of Sports Sciences, Polytechnic of Guarda, 6300-559 Guarda, Portugal; bartolomeu@ipg.pt; 2Department of Sport Sciences and Physical Education, Instituto Politécnico de Bragança, 5300-252 Bragança, Portugal; tatiana_sampaio30@hotmail.com (T.S.); morais.jorgestrela@gmail.com (J.E.M.); 3Research Center in Sports Sciences, Health and Human Development (CIDESD), 5000-801 Vila Real, Portugal; jpco-2001@live.com.pt; 4Department of Sports Sciences, University of Beira Interior, 6201-001 Covilhã, Portugal

**Keywords:** swimming performance, speed, dynamic balance, Y-balance test

## Abstract

Besides recurrently assessed water-based parameters, there are also some individual characteristics that affect swimming performance that are not water related. In the past few years, dynamic balance has been associated with land sports performance. Conversely, evidence on this topic in swimming is scarce. The purpose of this study was to assess the association between on-land dynamic balance and swimming performance. Sixteen young adults and recreational swimmers were recruited for the present study (8 males 20.8 ± 2.0 years, and 8 females 20.1 ± 1.9 years). A set of anthropometric features were measured. The upper quarter Y-balance test was selected as a dynamic balance outcome, and swimming speed as the swimming performance indicator. The results showed a moderate and positive correlation between dynamic balance and swimming performance (*p* < 0.05). Speed fluctuation was highly and negatively related to swimming speed (*p* < 0.001), i.e., swimmers who had higher scores in the dynamic balance were more likely to deliver better performances. This suggests that in recreational swimmers, the stability and mobility of the upper extremity had a greater influence on swimming performance. Therefore, swimming instructors are advised to include dynamic balance exercises in their land-based training sessions to improve their swimmers’ performance.

## 1. Introduction

The deterministic model for swimming performance shows that there is no single path to enhance performance. Rather, the interplay of several parameters determines swimming performance [1]. The underlying parameters affecting swim performance are not from a single scientific domain. Instead, they come from various domains, pointing out the need for an interdisciplinary approach to excel in swimming.

Besides task-related parameters (i.e., variables assessed during and related to actual swimming), organismic-related parameters (i.e., variables related to the swimmer itself) that are not specific to the swimming task have been indirectly correlated to swimming performance, e.g., dry-land upper-body strength [2,3], has been reported to be positively correlated with swimming speed. Anthropometric features, such as height [4,5], arm span [6,7], and hand surface area [8], have been reported to directly or indirectly affect swimming speed and other performance-related parameters [9,10,11].

The assessment of balance (a motor control outcome) by stabilometric techniques has been used for decades in sport sciences [12] as well as in rehabilitation programs [13]. Notwithstanding, dynamic balance has gained some traction in the past decade. Dynamic balance aims to examine the ability of a subject to perform a certain movement or combination of movements and return to baseline without losing control over it (i.e., keeping the balance). Field tests like the Star Excursion Balance Test (originally described by Gray [14]) or, more recently, the Y-balance test (originally proposed by Plisky et al. [15]) are used to assess dynamic balance. These tests are more cost-effective yet reliable in comparison to other techniques to monitor balance [15,16,17].

Although originally used to measure single-leg balance and reach distance, the Y-balance test was eventually adapted to assess the upper quarter balance [18,19], i.e., the subject’s ability to keep a plank position while reaching with one hand as far as possible in three directions. Dynamic balance and, notably, the upper quarter Y-balance Test (UQ-YBT) have been selected to monitor swimmers less than a handful of times as far as our understanding goes. Studies explored the differences in balance scores between athletes of different sports [20], competitive levels [21,22,23], and under different types of training regimes [24]. One study selected the UQ-YBT as a benchmark standard to validate another balance test in adolescent swimmers [25]. In addition to scarce literature, no records were found exploring hypothetical associations between upper-quarter dynamic balance and swimming performance.

Swimming speed is the parameter most often selected to assess performance in the sport of swimming. It depends, among others, on the ability of the swimmer to produce thrust, which in turn, is generated by the periodic action of the upper limbs [26,27]. Moreover, the arms’ actions are related to the swimmer’s motor control [1]. One can argue that the dynamic balance is an indicator of motor control. Hence, it is possible to speculate that a positive correlation between swimming speed and UQ-YBT might occur. Furthermore, one might wonder about interlimb differences in recreational swimmers. On the one hand, these swimmers lack an effective strength, flexibility, balance, or in-water technique training program. This might preserve the natural dominant/non-dominant interlimb asymmetries, also in dynamic balance. On the other hand, experienced and expert swimmers have shown asymmetries in various performance-related parameters [28,29,30,31,32]. The presence of asymmetries in the upper quarter dynamic balance remains unexplored and seems of importance given the literature stating that interlimb asymmetries might affect performance and/or increase injury likelihood. Thus, the aim of the present study was to assess the correlation of anthropometric features and the UQ-YBT with swimming performance, i.e., swimming speed. Further, interlimb differences were tested. It was hypothesized that larger anthropometric features and better UQ-YBT scores would be positively and significantly correlated with swimming performance.

## 2. Materials and Methods

The sample was composed of 16 young adults (8 males: 20.8 ± 2.0 years old; 8 females 20.1 ± 1.9 years old). They were all recreational swimmers with a previous background in swimming (4.1 ± 2.0 years) and were taking part in a swimming program twice a week for the 6 months prior to this study. All participants were clinically healthy, with reports of any musculoskeletal injury in the past six months. The tests took place on two consecutive days. On the first day, the participants performed the UQ-YBT, and the anthropometric parameters were measured. On the second day, they performed a 25 m all-out front-crawl bout. All procedures were in accordance with the Declaration of Helsinki regarding human research, and the participants provided informed consent. The Polytechnic Ethics Board approved the research design (No. 72/2022).

### 2.1. Dynamic Balance Measurements

According to the literature, there is a learning curve undergoing the UQ-YBT for the first time. Thus, subjects were asked to attend a demonstration of the procedure beforehand and to perform four practice trials as suggested elsewhere [33,34].

The UQ-YBT was performed according to the procedure described in detail in the literature [18]. Participants were asked to be in a push-up position with their feet shoulder-width apart. With one hand on the stand, the other was free to move and push the moving platform along the measuring tape: (i) in the medial direction, (ii) under the trunk in the inferolateral direction, and (iii) in the superolateral direction. Figure 1 depicts the UQ-YBT protocol. The test was performed three times with the dominant limb and then repeated three times with the non-dominant limb. This test sequence and number of repetitions were the same for both the right and left upper limbs for all participants. A trial would be deemed invalid and repeated if the subject was unable to keep the balance on the one hand on the platform (or either touched the floor with the reach hand or fell off the platform), forcefully pushed the moving platform away, used that platform for support, failed to return the reach hand to the starting position in a controlled manner, or lifted one foot off of the floor. To avoid bias, two researchers simultaneously and independently scored the trials that were averaged afterward. The maximum reached distances were then divided by the subject’s upper limb length, i.e., were relativized to allow between-subjects comparisons. For all participants, medial, inferolateral, and superolateral scores were averaged into a final composite score for the right hand (UQ-YBT_R) and the other for the left hand (UQ-YBT_L). Limb symmetry index (LSI) was calculated for the composite scores, as described in the literature [35]. The index was computed as the ratio of the dominant upper-limb composite score over the non-dominant upper-limb composite score. Furthermore, for all three directions of movement, the difference between limbs was calculated for: (i) the interlimb reach distance difference (AbsDif, in cm) and (ii) the interlimb reach distance difference after being relativized to the limb length (RelDif, in %).

### 2.2. Anthropometric Measurements

The body mass (BM, in kg) was measured on an electronic scale (Tanita, MC 780-P, Tokyo, Japan). The height (H, in cm) was measured by an electronic stadiometer (Seca, 242, Hamburg, Germany). The arm span (AS, in cm) and the hand surface area (HSA, in cm^2^) were measured by digital photogrammetry. To measure AS, swimmers were placed near a 2D calibration object in the upright position with both upper arms in lateral abduction at a 90° angle to the trunk. Both upper arms and fingers were fully extended. The distance between the tips of the third fingers was measured with a dedicated software program (UDruler v3.8, AVPSoft, Pittsburgh, PA, USA) [36]. For the measurement of the hand surface area (HSA), the swimmers’ palms were photographed with a digital camera (Sony a6000, Tokyo, Japan). Each HSA was calculated using a dedicated software program again (Udruler, AVPSoft, USA) [36].

### 2.3. Swimming Speed Measurement

Before data collection, swimmers underwent a standard pre-race warm-up protocol for sprint events, as suggested in the literature [37].

A nylon cable that unfolded over the trial from a speedometer device (SpeedRT, ApLab, Rome, Italy) and attached to a belt tied around the swimmer’s waist was used to measure swim speed. The speedometer transmitted the signal to software (Speed RT 2013, ApLab, Rome, Italy) that was streaming the swimmer´s speed in real-time [38]. After an auditory signal, the swimmer performed a push-off from the headwall without underwater kicking. To avoid possible large variability in swimming speed when breathing, swimmers were requested to perform non-breathing stroke cycles between the 10th and the 20th meters marks. To correctly identify the crossing of the vertex at the 10th and 20th-meter marks, a video camera (Hero 7, GoPro Inc., San Mateo, CA, USA) recorded the bout in the sagittal plane. To synchronize the speedometer and the camera, a light trigger in the recording field was turned on whenever the former began recording. The speedometer acquired data at a rate of 100 Hz, which was thereafter exported to a signal processing software (AcqKnowledge v3.9.0, Biopac Systems, Santa Barbara, CA, USA). After residual analysis, the signal was handled with a Butterworth 4th-order low-pass filter (cut-off: 5 Hz). The speed of the 10-m length was averaged. The intra-cyclic variation of the speed (speed fluctuation, dv) was also calculated from the speed-data signal according to the literature [39].

### 2.4. Statistics

Mean and one standard deviation were calculated as descriptive statistics. Shapiro–Wilk test was used to check data normality. Spearman’s correlation coefficient was selected and interpreted as: negligible if r_s_ < 0.3; low if 0.3 ≤ r_s_ < 0.5; moderate if 0.5 ≤ r_s_ < 0.7; high if 0.7 ≤ r_s_ < 0.9 and very high if 0.9 ≤ r_s_ ≤ 1 as suggested elsewhere [40]. Correlation agreements between both sexes (swimming speed vs. remaining variables) were computed by Fischer’s z-score [8]. Overall, non-significant differences (*p* < 0.05) were noted between correlations (only in one variable, i.e., medial reach for absolute and relative difference, a significant difference was noted), suggesting that both sexes could be pooled together.

## 3. Results

Mean ± 1SD for anthropometrics, balance, and swimming are presented in Table 1. In Table 2 can be seen that there was a moderate and positive correlation between swimming speed and both balance variables: YBT_R (r_s_ = 0.568, *p* = 0.027) and YBT_L (r_s_ = 0.539, *p* = 0.038). Furthermore, there was a negative and high correlation between swimming speed and dv (r_s_ = −0.804, *p* < 0.001) (Table 2). Composite scores (YBT_R and YBT_L) were correlated to each other (r_s_ = 0.954, *p* < 0.001) (Table 2). The LSI was higher than the standard 90% cut-off value (Table 1).

## 4. Discussion

The aim of the present study was to assess the association between anthropometric features and UQ-YBT with swimming performance. The main findings point out that swimming speed in young adult recreational swimmers was positively and significantly correlated with UQ-YBT and negatively and significantly associated with dv (i.e., less dv led to fastest swimming speeds). 

The dynamic balance scores have been used to screen deficits in neuromuscular control due to pathologies [41,42,43,44], injury prediction [45,46,47], the effect of a certain treatment [44,48,49,50], or rehabilitation/physical activity program [42,51,52,53,54] in patients with various diseases and return to sport readiness [55,56]. Furthermore, the performance delivered in these tests has also been positively related to physical fitness [57,58] and sports performance [22,23,59]. The present results are in tandem with the literature, as higher scores in the UQ-YBT (better upper quarter dynamic balance) were correlated to faster swimming speeds (better performance). Butler et al. [23] found that professional players scored better in dynamic balance than their collegiate or high school counterparts. Moreover, both González-Hernanez et al. [59] and Brumitt et al. [22] reported better scores amongst athletes in higher competitive levels and differences in dynamic balance across field positions in soccer and volleyball players, respectively. The authors put forward that those differences in scores can be explained by the training programs implemented in the teams rather than the participants´ competitive level. Thus, one could speculate whether swimming specialization would affect the association between speed and dynamic balance. This topic needs further investigation. On the other hand, this shows the need for upper-quarter dynamic balance training even for low-tier swimmers, such as recreational-level swimmers. Despite the positive correlation, one could only wonder if the results from the UQ-YBT of these recreational swimmers are within the expected values for their expertise level. This topic also requires further investigation, as Brumit et al. [22] stated the need to make available normative data for different levels of expertise. This information would help coaches to have a better insight into the swimmers´ dynamic balance, where there is room for improvement, and if there is the need to prescribe more drills to improve dynamic balance on land or in water.

Sports performance is a multifactorial phenomenon. Numerous factors from different scientific domains interplay in a web-like structure where one factor affects another in a cascade fashion, ultimately affecting the main outcome (the performance). Examples of parameters found in this complex and dynamic system of interactions are the swimmer´s height and arm span [1]. However, being taller or having a wider arm span is not a guarantee of delivering a better swimming performance. Instead, these influence other parameters such as body volume, body mass, stroke length, and propelling efficiency [1] that, in turn, determine other variables. The same line of reasoning could be applied in the present study, as none of the anthropometric variables measured had a significant association with swimming speed. Swimming is a sport that requires a unique motor control refinement to enhance performance. Marginal improvements in the technique lead to better performances [60], and athletes’ movement strategies may be different depending on their competitive level. Recreational swimmers may not have yet acquired the finest technique, which would enable them to unlock the potential from unique anthropometric parameters and, thus, possibly, the non-significant correlation.

Swimming speed is the net balance between the thrust produced and the drag acting against the body. Such balance between these external forces varies within and between each stroke cycle leading to instant changes in swimming speed. Better motor control, i.e., greater regularity in the arms and legs’ actions during propulsion and better body alignment (better hydrodynamics), are related to a decrease in this intracyclic variation of the horizontal speed of the swimmer (i.e., speed fluctuation). Thus, less speed fluctuation (dv) is correlated to better performances [5,61]. In the present study, speed was negatively related to dv, which is in line with the literature. Bartolomeu et al. [61] reported an inverse correlation of dv with swimming speed for all four competitive swimming strokes. Barbosa et al. [5] reported a decrease in dv with increasing expertise. The values observed for the recreational swimmers in the present study are in tandem with the literature, as dv was higher than that observed for all regional competitors, national competitors, and national record holders in a study by Barbosa et al. [5]. Despite the fact that better motor control is needed to improve dv, in the present study, the UQ-YBT scores were not correlated to swimming speed. Once more, the swimmers’ level could be the cause. Perhaps all swimmers had small hydrodynamic balance expertise in water, leading to a higher, nonlinear, and thus non-significant correlation.

Some concerns have been raised in the past couple of years regarding the methodology of dv calculation based on the coefficient of variation (CV), as performed in the present study [62,63]. Indeed, mixed findings can be found in the literature regarding the relationship between dv based on CV and swimming speed [64,65]. CV is dependent on the mean velocity and its standard deviation. Some authors have pointed out that both variables might not vary the same way throughout the swimming bout. Gonjo et al. [63] stated that there is evidence for maintenance in the difference between the intra-cycle maximum and minimum velocity (standard deviation) despite an observable decrease in the mean velocity. In such a case, CV would be biased by the mean velocity. However, to keep the standard deviation of the mean speed stable, swimmers need to have a stable inter-limb coordination pattern, a condition that is more prone to be observable in highly experienced swimmers, which can, to a certain extent, maintain the technique despite fatigue. In learning to swim and recreational programs, one of the main focuses is to reduce the large intracyclic speed fluctuations due to poor technique (namely interlimb coordination). Several studies have reported larger dv in less skilled swimmers [5,65,66]. The present results are in tandem, as those with the higher dv were indeed the slowest, thus, less skilled swimmers. As an example, in a study comparing performance among levels of expertise, Barbosa et al. [66] reported values of dv of 18.40 ± 6.00 for non-expert swimmers (the higher value among all groups), being these subjects described as those who practice the sport at a non-competitive level on a regular basis, thus, just like the recreational swimmers in the present group. Possibly the swimmers from our study have less experience in swimming (the years of experience in the aforementioned study are not reported), explaining the larger values obtained in the present study (29.71 ± 13.37). Whichever path dv-related research takes, its calculation, as performed in the present study, appears to continue as a good proxy of the speed fluctuation, at least for the non-elite swimmers.

The body of knowledge in motor control points out consistent interlimb differences. Interlimb differences have been reported in soccer [67,68], volleyball [69], basketball [69,70], and running [71] for several parameters such as force production [72], power output [72], range of motion [73], jump height or length [74,75], and dynamic balance [45]. In swimmers, interlimb differences have been observed in dry-land strength [28], power output [29,30], hand and feet force [31], and arm kinematics [32]. The literature reports a cut-off value of 10–15% when assessing interlimb differences. Differences over cut-off values denote a larger likelihood of musculoskeletal injuries and poorer sports performances. Although having a reference value can serve as a guide to coaches and physiotherapists, swimming researchers have reported asymmetries above 10–15% in swimming-specific tasks in elite swimmers [32,76]. Seifert et al. [77] concluded that asymmetries in arm coordination might be due to different roles between limbs (propulsion and rhythm for the dominant upper arm vs. propulsion and compensation for breathing laterality for the non-dominant upper arm). Bartolomeu et al. [78] have hypothesized that swimming is probably a task that benefits up to a certain extent, from some asymmetry, possibly above 10%. In the present study, the mean LSI was about 96%. The fact that lower symmetry values are often found in parameters measured during actual swimming (power output or hand/feet force production) when compared to those of land-based measurements, such as the dynamic balance of the present study and others [71,72,79], confirms such a hypothesis. Notwithstanding, more research is needed studying land- and water-based asymmetries in the same subjects. 

In the present study, LSI was not correlated to any of the variables under study. In tandem, neither the absolute difference nor the relative difference between the dominant and non-dominant arm was related to any other variable in the three reaching directions. The high symmetry in interlimb dynamic balance (mean ~96%) can possibly explain the absence of a significant correlation between inter-limb symmetries and speed. Indeed, literature has reported the same lack of meaningful correlation. Power output and hand force symmetries were not correlated to performance in tethered swimming [30] and free swimming [78], even when high asymmetries were present. The reason behind these findings can, yet again, be underpinned by the different roles of both limbs. Under this hypothesis, one might argue that having more or less symmetry would not matter (up to a certain extent) as this would be a reflection of the individual technique and the individual contribution of each limb to that technique.

Regarding dynamic balance, there are a few studies that correlated symmetry levels (rather than asymmetry) with injury risk. Plisky et al. [45] reported that composite scores of a dynamic balance test lower than 94% would indicate an increased injury risk in basketball players. Butler et al. [80] indicated that football players who scored below 89.6% on the test were 3.5 times more likely to get injured. Recreational swimmers of the present study scored an average of 87.10% and 88.42% for the YBT_R and YBT_L, respectively, which supposedly means an increased injury risk. However, the subjects recruited are not under intense training protocols and reported no injury history. Thus, the low scores are due to a lack of core stability more than structural imbalances. This suggests that the cut-off values proposed by the literature might be unsuitable for recreational swimmers or for swimming parameters at all. The rationale that injury risk cut-off values for interlimb asymmetries might be task-specific has been put forward before for land-based activities [72,81,82,83]. Along the same line, the findings from the present study suggest that the same phenomenon might occur for injury risk prediction from the dynamic balance results in swimmers. However, further analysis should be carried out as the design selected and analysis performed in the present study cannot confirm such an assumption. Nevertheless, the amount of research on the influence of interlimb symmetries on injury risk and performance showcases its importance. In swimming, given such between-subject differences regarding symmetry [30,31], customized assessments and interventions may be needed to address specific interlimb symmetries and injury risks. Asymmetries can occur naturally or develop over time due to training, technique, or anatomical factors. Lower levels of symmetry in swimming can come from task-specific factors, for example, poor technique, or from other non-specific factors, such as musculoskeletal imbalances. In a study by Evershed et al. [28], it was reported that all swimmers who presented muscular symmetry on dry-land tests presented a symmetry in force production while swimming. From those who presented dry-land strength asymmetries, some used compensatory movements and presented symmetry in force production, and others maintained the force asymmetry. The fact that all participants were national-level swimmers supports the idea of a high inter-individual difference. Thus, the need for individual assessment and the exercise of special caution when using typical cut-off values for injury risk assessments. Nevertheless, the present values might act as a reference for healthy recreational swimmers for the dynamic balance scores and symmetry.

Despite the fact that the statistical tests used are robust for small samples, the number of participants recruited can be addressed as a limitation in the present study; thus, results should be interpreted with caution. Different swimming distances impose variations on stroke frequency x stroke length combinations. Hence, other distances and other submaximal swimming velocities should be tested in the future. Further investigation should also be conducted into the role of dynamic balance on performance among swimmers of other competitive levels and on stroke-specialized swimmers. Furthermore, dynamic balance normative data for different levels of expertise may need to be established. These values could provide insight into the need for improvement to maintain performance and as the benchmark to return to that competing level after a musculoskeletal injury. 

In summary, speed fluctuation and upper quarter dynamic balance of recreational swimmers were significantly associated with swimming speed. Swimming coaches are advised to incorporate on-land dynamic balance exercises and/or pay more attention to in-water dynamic balance drills in recreational swimmers. This improvement in dynamic balance training could also improve swimming technique which may, in turn, unlock the potential of other performance-related determinants such as anthropometrics.

## 5. Conclusions

In recreational swimmers, better upper quarter dynamic balance and lower dv were associated with faster swimming speeds (YBT_R: r_s_ = 0.568, YBT_L: r_s_ = 0.539, and dv: r_s_ = −0.804). Recreational swimmers showed high upper quarter dynamic balance symmetry (LSI: 96.03 ± 2.57%). Dynamic balance training, either specific or embedded in regular training sessions, should not be disregarded, given the observed correlation between upper-quarter dynamic balance and performance.

## Figures and Tables

**Figure 1 jfmk-08-00096-f001:**
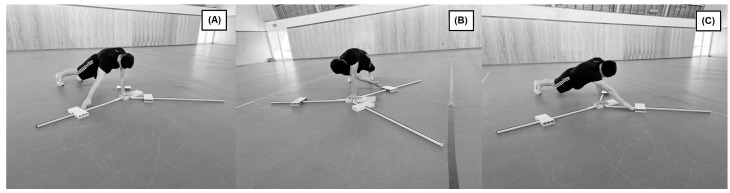
Protocol of the UQ-YBT. Panel (**A**)—medial reach; Panel (**B**)—inferolateral reach; Panel (**C**)—superolateral reach.

**Table 1 jfmk-08-00096-t001:** Descriptive statistics (mean ± 1SD) for all variables in the study.

	Mean ± 1SD
BM [kg]	71.67 ± 7.81
H [cm]	173.20 ± 7.67
AS [cm]	173.13 ± 7.44
HSA_R [cm^2^]	130.90 ± 121.02
HSA_L [cm^2^]	123.81 ± 12.49
YBT_R [%]	87.10 ± 12.88
YBT_L [%]	88.42 ± 11.32
LSI [%]	96.03 ± 2.57
AbsDif_medial [cm]	5.30 ± 3.24
AbsDif_inferolateral [cm]	5.57 ± 3.53
AbsDif_superolateral [cm]	4.40 ± 3.94
RelDif_medial [%]	5.97 ± 3.85
RelDif_inferolateral [%]	6.44 ± 3.39
RelDif_superolateral [%]	5.17 ± 4.27
v [m∙s^−1^]	1.21 ±1.17
dv [%]	29.71 ± 13.37

BM—Body mass; H—Height; AS—Arm span; HSA_R—Right-hand surface area; HSA_L—Left-hand surface area; YBT_R—Score for the right arm in the UQ-YBT; YBT_L—Score for the left arm in the UQ-YBT; v—Swimming speed; dv—intra-cyclic speed fluctuation.

**Table 2 jfmk-08-00096-t002:** Spearman correlation coefficient between swimming speed and all other variables in study.

	Speed [m∙s^−1^]	
	r_s_	*p* Value
BM [kg]	−0.063	0.823
H [cm]	0.437	0.103
AS [cm]	0.362	0.185
HSA_R [cm^2^]	0.146	0.603
HSA_L [cm^2^]	0.496	0.060
YBT_R [%]	0.568	0.027
YBT_L [%]	0.539	0.038
LSI [%]	−0.124	0.520
AbsDif_medial	−0.007	0.980
AbsDif_inferolateral	0.168	0.551
AbsDif_superolateral	−0.031	0.914
RelDif_medial	−0.046	0.869
RelDif_inferolateral	0.032	0.909
RelDif_superolateral	−0.236	0.398
dv [%]	−0.804	<0.001

BM—Body mass; H—Height; AS—Arm span; HSA_R—Right-hand surface area; HSA_L—Left-hand surface area; YBT_R—Score for the right arm in the UQ-YBT; YBT_L—Score for the left arm in the UQ-YBT; dv—speed fluctuation.

## Data Availability

Nothing to declare.

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
