# Peer review of "Association between the Upper Quarter Dynamic Balance, Anthropometrics, Kinematics, and Swimming Speed"

_jfmk, 2023, doi:10.3390/jfmk8030096_

Round 1
Reviewer 1 Report
Thanks a lot for giving me the opportunity to review this very interesting manuscript. I think that the authors did a great job in conducting the study. I would just like to point out some issues that need revision:
- Wrong conclusion: While swimmers who had higher scores in dynamic balance were more likely to deliver better performances, from the presented data it cannot be deducted that dynamic balance exercises will improve swimming performance. It first needs to be proven that this relationship also exists.
- Statistics: Did the authors try to transform the data before running the non-parametric test? Which transformations have been tried? If not, please try and then use a parametric test, if the data become normally distributed.
- Please add figures that show the data and correlation of swimming speed with the balance variables.
- Hand surface area is HSA - please correct in line 132.
The English language needs to be improved, as there are lots of mistakes in the manuscript. Examples:
- water-related needs a dash (many dashes missing in the manuscript)
- Line 56: as far as our understand goes (understanding)
- Line 56: in one hand (On the one hand)
I recommend to use Deepl or a professional language editing service to improve the writing, as there a many such errors in the text.
Author Response
Kindly refer to attached file.

Reviewer 2 Report
The evaluated article is very interesting, having as objective: Assess the correlation between the on-land dynamic balance and swimming performance.
Its authors are recommended to solve the following approaches:
1) Place the values of the Spearman's Test in the abstract section.
2) Based on the non-parametric value of Spearman's, describe in the methods section how the data normality was determined.
3) The sample studied is very small (16 swimmers). In this sense, the existence of care in the results achieved must be specified as a research limitation, recommending expanding the studies to representative samples.
4) In the methods section, the sample is classified by gender (Lines: 81-82); however, the linear correlations carried out do not classify the results according to the swimmer sex, studying the results as a whole. Considering the notable differences existing in various variables of sports performance according to gender, the article authors of are asked to justify whether or not said unification could have an influence on the research results. On this aspect, it is recommended to keep in mind studies that classify the results by gender in the future, determining their particularities.
5) In the conclusions section, it is recommended to describe the most relevant quantitative results that support the qualitative approaches made.
Evaluate with a specialist
Author Response
Kindly refer to attached file.

Round 2
Reviewer 1 Report
Thanks for revising the manuscript. I still believe that a figure would have been very beneficial, but that is in the end the author's choice.
Okay